# Hydrothermal Synthesis of Co_3_O_4_/ZnO Hybrid Nanoparticles for Triethylamine Detection

**DOI:** 10.3390/nano9111599

**Published:** 2019-11-11

**Authors:** Yanqiong Yang, Xiaodong Wang, Guiyun Yi, Huimin Li, Chuang Shi, Guang Sun, Zhanying Zhang

**Affiliations:** 1School of Materials Science and Engineering, Henan Polytechnic University, Jiaozuo 454000, Chinamcsunguang@163.com (G.S.); zhangzy@hpu.edu.cn (Z.Z.); 2College of Chemistry and Chemical Engineering, Henan Polytechnic University, Jiaozuo 454000, China

**Keywords:** Co_3_O_4_-ZnO, nanoparticles, hydrothermal, triethylamine detection, gas sensor

## Abstract

Development of high performances gas sensors to monitor and detect the volatile organic compound triethylamine is of paramount importance for health and environmental protection. The Co_3_O_4_-ZnO nanoparticles composite was successfully synthesized by the one-step hydrothermal route and annealing process in this work. The gas sensitivity test results show that the composite exhibits excellent triethylamine-sensing performance at a cobalt content of 1 at%, indicating potential application for triethylamine detection. The sensor based on the Co_3_O_4_-ZnO composite had higher sensitivity to triethylamine, better selectivity, and faster response recovery rate compared with pure ZnO sensor. Combined with the structural characteristics of the characterized Co_3_O_4_-ZnO nanocomposite, the optimized triethylamine sensing performances can be ascribed to the p-n heterojunction effect between Co_3_O_4_ and ZnO, as well as the catalytic and high oxygen adsorption properties of Co_3_O_4_.

## 1. Introduction

Volatile organic compounds (VOCs) in industrial production and residential life are potential threats to air pollution [1]. As a typical colorless transparent VOC, triethylamine (TEA) is widely used in organic solvents, catalysts, preservatives, and synthetic dyes [2,3]. However, TEA is toxic, flammable, and explosive, and its leakage poses a serious threat to human health and environmental safety [4,5]. Chromatography, colorimetry, conductive polymer sensors, and luminescent gas sensors are several reported detection methods that limit the practical application of TEA detection due to their high cost and time-consuming operation [6]. Therefore, there is an urgent need for a method that is easy to fabricate and convenient for monitoring and detecting TEA.

Metal oxide semiconductor (MOS)-based gas sensors have been used extensively to detect hazardous gases because of their low cost, small size, high preparation flexibility, and the ability to detect multiple gases [7,8]. The typical n-type semiconductor ZnO is considered to be an extremely beneficial gas-sensing material by reason of good chemical stability, non-toxicity, high surface-to-volume ratio, suitable doping, and low cost [9,10,11]. It is worth noting that the fabrication of heterostructure composite sensors can enhance gas-sensing performances and significantly improve the low sensitivity and poor selectivity of pure ZnO materials [12,13,14]. Kim et al. [15] reported the CuO-ZnO heterostructure nanorods sensor to enhance H_2_S gas response and repeatability over pure ZnO. Zou et al. [16] designed a TiO_2_/ZnO-based NO_2_ gas sensor that can improve the gas sensitivity of ZnO. Gholami et al. [17] showed an enhanced ethanol response by adding a small amount of In_2_O_3_ in ZnO to build an In_2_O_3_-ZnO nanostructured gas sensor. Xu et al. [18] reported that NiO-ZnO heterojunction nanotubes sensor exhibit better selectivity for H_2_S than pure ZnO.

Moreover, Co_3_O_4_ is a representative p-type MOS with catalytic and high oxygen adsorption properties and is an ideal composite material for ZnO [19,20,21]. Zhang et al. [22] reported that the mesoporous Co_3_O_4_-ZnO microspheres sensor has an ethanol response about five times that of pure ZnO. Park et al. [23] pointed out that the Co_3_O_4_-ZnO composite nanoparticles network sensor differs from pure ZnO in that it has the advantage of significantly enhanced NO_2_ gas-sensing performance. Xu et al. [24] reported that a flytrap-like Co_3_O_4_-ZnO composite sensor had an eight-fold higher response to 100 ppm ethanol than pure ZnO. However, it has rarely been reported that Co_3_O_4_-ZnO composites are effective for TEA gas-sensing.

Here, the Co_3_O_4_-ZnO nanoparticles composite was designed and synthesized by a simple hydrothermal method, and its gas-sensing properties were further studied. Attractively, the sensor based on the Co_3_O_4_-ZnO composite exhibits higher sensitivity, better selectivity, and faster response recovery rate to TEA gas than pure ZnO, indicating that the Co_3_O_4_-ZnO nanoparticles composite is a promising candidate for TEA detection. In addition, the possible sensitization mechanisms of the Co_3_O_4_-ZnO composite were also discussed.

## 2. Materials and Methods

### 2.1. Preparation of ZnO and Co_3_O_4_-ZnO Nanoparticles

The reagents involved in the experiment included zinc acetate dihydrate (Zn(CH_3_COO)_2_·2H_2_O, 99%), cobalt acetate tetrahydrate (Co(CH_3_COO)_2_·4H_2_O, 99.5%), polyvinylpyrollidone (PVP), ethanol (C_2_H_5_OH), and deionized water. This experimental protocol modified the reported method [25]. A total of 0.439 g Zn(CH_3_COO)_2_·2H_2_O and 0.5 g PVP was added to 70 mL C_2_H_5_OH. After magnetic stirring for 30 min, 0.005 g (1 at%) Co(CH_3_COO)_2_·4H_2_O was added, stirring was continued for 30 min and sonicated for 10 min. The resulting mixture was then shifted to the 100 mL reaction vessel and heated to 180 °C for 9 h. After the reaction was completed, it was centrifuged and washed alternately with C_2_H_5_OH and deionized water four times to obtain the precipitate. The above product, after drying in a 60 °C oven, was put in muffle furnace and treated at 600 °C for 2 h in an air atmosphere to obtain the stable sample. A series of samples were prepared by simply adjusting the content of Co, in which the molar ratio of Co/Zn was 0, 0.5, 1, 3 at%, respectively, and they were named ZnO, Co_3_O_4_/ZnO-1, Co_3_O_4_/ZnO-2, Co_3_O_4_/ZnO-3, respectively.

### 2.2. Characterizations

The study of the crystal structure of the sample was carried out on the powder X-ray diffraction (XRD, Bruker-AXS D8, Bruker, Madison, WI, USA) under Cu-Kα radiation (40 kV, 25 mA). Field emission scanning electron microscopy (FESEM, Quanta 250 FEG-SEM, FEI, Eindhoven, Netherlands) and transmission electron microscopy (TEM, JEOL JEM-2100, Tokyo, Japan) were used to investigate the morphology of the sample. In addition, the energy dispersion spectrum (EDS, INCAENERGY 250, FEI, Eindhoven, Netherlands) characterized the elemental composition. The Brunauer–Emmett–Teller (BET, Quantachrome, Boynton Beach, FL, USA) tested the specific surface area. X-ray photoelectron spectroscopy (XPS, ThermoFischer ESCALAB 250Xi, MA, USA) was further used to analyze the chemical valence of the sample, and C1s = 284.80 eV was its charge correction standard.

### 2.3. Construction of Gas Sensors

The paste obtained by mixing a small amount of sample and an appropriate amount of deionized water was brushed on the Ag-Pd electrode region of the purchased ceramic substrate (electrode line width 0.2 mm, pitch 1.0 mm; entire substrate 13.4 × 7 mm) (Beijing Elite Tech Co., Ltd., Beijing, China) to obtain a gas sensor (Figure 1), which was aged in a 60 °C oven for one night to improve stability. The manufactured sensors were placed on the CGS-4TPS intelligent sensing analysis system for sensitivity testing, and the relative humidity condition was basically maintained at 30%. The ratio of the resistance R_a_ in the air to the resistance R_g_ in the test gas was designated as the response value, and the time at which the sensor reached the maximum response of 90% and 10% was expressed as the response and recovery time, respectively. In particular, pairs of adjustable probes were in contact with and were attached to both ends of the fabricated sensor. The power supply, water circulation system, temperature control system, and data acquisition system were then turned on. Initially in the atmosphere, and when the resistance of the sensor collected on the display was relatively stable, the target gas was introduced into the test chamber after an interval of 20 s, after which the test chamber was quickly closed. In the data acquisition system, the relatively-stable resistance when the sensor is in the air was set to R_a_; R_g_ is the resistance of the sensor collected by the system in the test gas, and R_a_/R_g_ is selected in the sensor response curve type column. After the target gas stayed in the test chamber for about 240 s, the lid on the upper side of the test chamber was opened to release the test gas. When the sensor’s response value basically returned to 10% of the maximum response, it was clicked in order to stop collecting data and to save the data. In this way, the R_a_/R_g_ value was directly recorded in the data acquisition system on the intelligent sensing analysis system, and finally exported by computer.

## 3. Results

### 3.1. Materials Characterizations

The XRD patterns of each sample are shown in Figure 2. In Figure 2a–d, almost all of the diffraction peaks match the hexagonal wurtzite structure ZnO (JCPDS no. 89-0511) with a lattice constant of a = b = 3.249 Å and c = 5.205 Å. The three strong peaks are matched with the (101), (100), and (002) crystal faces of ZnO in order of strength from strong to weak. Further, as shown in Figure 2, Figure 2f is an enlarged view of a shaded portion of yellow in Figure 2a–d. Figure 2e shows that, in the range of 2θ = 30°~40°, 2θ = 36.263° in Figure 2a is very close to the diffraction angle of 2θ = 36.26° corresponding to the (101) crystal plane of ZnO (JCPDS no. 89-0511). Moreover, the peak positions of Figure 2b–d are offset by 0.02°, 0.04°, 0.06°, respectively, compared to Figure 2a. This may be mainly due to the introduction of Co_3_O_4_ in the composite. However, the sample with a Co content of 3 at% in Figure 2d still has no obvious characteristic diffraction peak of Co_3_O_4_, which may be caused by the low content of Co in the composite sample.

Figure 3a,b shows SEM images of ZnO and Co_3_O_4_/ZnO-2 samples, respectively. As shown in Figure 3a, the ZnO sample consists of agglomerated nanoparticles lumps. With the introduction of Co in the sample, the Co_3_O_4_/ZnO-2 sample in Figure 3b shows nanoparticles lumps with smaller average size and denser distribution. It is further shown that the composite sample has a larger available surface area and is advantageous for adsorption of the target gas. Figure 3c,d shows TEM and HRTEM images of the Co_3_O_4_/ZnO-2 sample, respectively. It can be seen from Figure 3c that Co_3_O_4_/ZnO-2 has an average particle size of about 30 nm. The HRTEM image in Figure 3d has lattice fringes with different spacings of 0.26 and 0.23 nm, which are in good agreement with the (002) crystal plane of ZnO and the (222) crystal plane of Co_3_O_4_, respectively. The EDS of the Co_3_O_4_/ZnO-2 sample shown in Figure 3e–g contains Zn, O, and Co elements. It is indicated that the successful introduction of Co in the composite sample can also echo the HRTEM result.

The pore structures of ZnO and Co_3_O_4_/ZnO-2 were tested by N_2_ adsorption–desorption instrument. The type IV adsorption isotherm results in Figure 4a indicate that the two samples have the mesoporous structure. Figure 4b shows that the pores of the sample are mainly about 16 nm. The Co_3_O_4_/ZnO-2 sample still has pores in the range of 32 nm, and even the maximum pore diameter is about 36 nm. These larger pores are favorable for gas adsorption and diffusion. At the same time, the calculated BET surface areas of ZnO and Co_3_O_4_/ZnO-2 were 17.733 and 21.150 m^2^g^−1^, respectively. The larger specific surface area and pore size of Co_3_O_4_/ZnO-2 means that there are more available areas to facilitate gas adsorption and surface reaction, which will help its gas sensitivity.

In order to further analyze the structural surface state, the samples were subjected to XPS measurement. The full spectra analyses of ZnO and Co_3_O_4_/ZnO-2 are shown in Figure 5a. The peak positions of the two samples are basically the same and both have characteristic peaks of Zn and O elements, except that we can see that there is a weak Co peak in the Co_3_O_4_/ZnO-2. Figure 5b shows the narrow-area scan of Co 2p in the Co_3_O_4_/ZnO-2 sample, where the two main peaks at 780.57 and 796.75 eV are consistent with Co 2p_3/2_ and Co 2p_1/2_, respectively. Besides, there are two vibrating satellite peaks at 784.61 and 802.81 eV, which further confirm that the composite sample forms the Co_3_O_4_ crystal phase [25,26]. Figure 5c,d shows the high resolution spectra of O 1s in ZnO and Co_3_O_4_/ZnO-2 samples, respectively. Both samples were divided into three oxygen peaks, indicating the presence of three different oxygen species in each of the two samples. Figure 5c shows that the peaks of three different O 1s states in ZnO are located at the binding energies of 529.85, 530.83, and 531.93 eV, wherein the contents of lattice oxygen (O_L_), vacancy oxygen (O_V_), and chemisorbed oxygen (O_C_) species are 66.8%, 22.99%, and 10.21%, respectively. Figure 5d shows that the peaks of three different O 1s states in the Co_3_O_4_/ZnO-2 samples are at 529.97, 530.95, and 531.84 eV, with O_L_, O_V_, and O_C_ contents of 66.6%, 16.75%, and 16.65%, respectively. Table 1 shows the states of O_L_, O_V_, and O_C_ in ZnO and Co_3_O_4_/ZnO-2. The results show that the surface of the Co_3_O_4_/ZnO-2 composite sample has more chemisorbed oxygen, which may be closely related to the high oxygen adsorption characteristic of the Co_3_O_4_. More O_C_ on the sensing material promotes redox reaction with the target gas, meaning the Co_3_O_4_/ZnO-2 sensor may have higher sensitivity than pure ZnO.

### 3.2. Gas-Sensing Performances

In view of the fact that the working temperature has a major impact on sensitive materials, 200 ppm TEA was initially tested at different temperatures. In Figure 6a, as the working temperature increases from 250 to 320 °C, the responses of all sensors display the similar upward convex shape and maximum response at 280 °C. The relationship between temperature and response of the sensor can be attributed to the kinetic and thermodynamic behavior of gas. When the working temperature is lower than 280 °C, there is not enough thermal energy to activate the TEA molecule to completely react with the oxygen ions, so the response of the sensor gradually increases as the temperature rises. When exceeds 280 °C, the gas behavior may be biased toward the desorption side, which reduces the utilization of the sensing material, resulting in a reduced response. At 280 °C, the TEA molecule has the highest oxidation rate and the sensor achieves the highest response. Therefore, 280 °C was determined to be optimum working temperature and applied to the following tests. In addition, Co_3_O_4_-ZnO composite sensors can show higher responses than pure ZnO sensor at different working temperatures, indicating that the ZnO sensor modified by Co_3_O_4_ can significantly improve the TEA sensing performance of pure ZnO. Meanwhile, the Co_3_O_4_/ZnO-2 sensor showed the highest response in three different composite sensors, indicating that the optimum Co-doping amount of the composite was 1 at%.

Concentration-response correlation tests were performed on two representative ZnO and Co_3_O_4_/ZnO-2 sensors. Figure 6b shows that the responses of both sensors are positively correlated with TEA concentration and the composite sensor has a higher response. Notably, the Co_3_O_4_/ZnO-2 response has a larger upward tilt angle in the lower TEA concentration range (10–200 ppm), which is more suitable for practical, low concentration TEA detection applications. As shown in Figure 6c, as the TEA concentration increased from 10 to 1000 ppm, the responses of sensors gradually increased. Figure 6d shows that the measured response/recovery times for ZnO and Co_3_O_4_/ZnO-2 are 76/45 and 25/36 s, respectively, indicating that the Co_3_O_4_/ZnO-2 sensor has a faster response–recovery rate.

Moreover, selectivity is also a key indicator for assessing the quality of gas sensors [27]. Figure 6e shows the selective testing of ZnO and Co_3_O_4_/ZnO-2 sensors for 200 ppm of different gases (TEA, formaldehyde, acetone, ethanol, and methanol). The experimental results show that the two sensors have the highest responses to TEA compared to other gases. The Co_3_O_4_/ZnO-2 sensor has a response (282.3) to 200 ppm TEA of approximately 12.8 times that of ZnO. In addition, the Co_3_O_4_/ZnO-2 composite sensor’s response to TEA is 20.3, 24.4, 20.5, and 49.7 times that of other gases, respectively, while ZnO is only 4, 6.5, 8.2, and 11.2 times. The excellent TEA selectivity of the Co_3_O_4_/ZnO-2 sensor may be due to the catalytic action of Co_3_O_4_ and the lower bond energy of the C–N in the TEA molecule compared with C=O, C–C, and O–H [26,28]. The results of TEA sensing properties studies for the different materials shown in Table 2 indicate that the Co_3_O_4_/ZnO-2 nanoparticles composite sensor constructed by this work has a lower working temperature and higher response.

The repeatability test for the Co_3_O_4_/ZnO-2 sensor is shown in Figure 6f. The result shows that the response of the Co_3_O_4_/ZnO-2 sensor during the repeated cycle test within 10 days only fluctuates within a very small range (the coefficient of variation of the responses is about 1.42%), further indicating that the Co_3_O_4_/ZnO-2 composite sensor prepared in this work has good durability for TEA detection.

### 3.3. Gas-Sensing Mechanisms

The sensing mechanism of n-type ZnO is mainly the regulation of the depletion layer [40,41,42,43]. The energy band diagrams of ZnO in air and TEA are shown in Figure 7a,b, respectively. In air, oxygen molecules are physically adsorbed on the surface of the material and become chemisorbed oxygen ions (O_2_^−^, O^−^ and O^2−^) by trapping electrons in the ZnO conduction band. Thence, the formation of an electron depletion layer in the surface region of ZnO results in an increase in sensor resistance. After introduction of the TEA, the TEA is oxidized by the active oxygen ions adsorbed on the material to compensate for the carrier (electron) concentration of ZnO. In this way, the measured resistance of the sensor is reduced by the narrowing of the material electron depletion layer.

In Co_3_O_4_-ZnO composite nanostructures, ZnO has a higher Fermi level than Co_3_O_4_, electrons will transfer from ZnO to Co_3_O_4_, and holes will shift from the opposite direction until Fermi energy level of balance [44,45]. Finally, a barrier (p-n heterojunction) is formed in the interface region [46,47,48]. Figure 7c,d are the energy band diagrams of Co_3_O_4_-ZnO in air and TEA, respectively. In the air atmosphere, the exposed surfaces of ZnO and Co_3_O_4_ form the electrons depletion layer and the holes accumulation layer due to the adsorption of oxygen species, respectively, and greatly promote the electron transfer process. Thus, the composite material forms a thicker depletion layer, leading to a higher resistance state of the sensor. Once the sensor is in the TEA atmosphere, the electrons released by the surface reaction will replenish the electrons concentration and neutralize the holes. This not only thins the depletion layer of the Co_3_O_4_-ZnO composite but also lowers the barrier height at the interface. Thereby, the resistance of the Co_3_O_4_-ZnO sensor drops sharply.

Apart from the p-n heterojunction effect between Co_3_O_4_ and ZnO, the catalytic and high oxygen adsorption characteristics of Co_3_O_4_ are also the main role of the Co_3_O_4_-ZnO composite in enhancing gas sensitivity. On the one hand, the catalytic properties of Co_3_O_4_ reduce the activation energy of TEA and the oxygen ion reaction. On the other hand, the presence of Co^2+^, which is easily oxidized in the Co^2+^/Co^3+^ system of Co_3_O_4_, promotes the adsorption of oxygen molecules on Co_3_O_4_-ZnO. The results accelerate the reaction between active TEA and abundant chemisorbed oxygen ions. In addition, due to the introduction of Co_3_O_4_, the Co_3_O_4_-ZnO material forms a unique, porous, nanoparticle-dense lumps structure with a smaller average size, resulting in a larger available specific surface area, providing more adsorption sites to facilitate surface reactions. Combined with these factors, the resistance of the Co_3_O_4_-ZnO gas sensor in air is greatly increased compared with the pure ZnO gas sensor, and further reduced in the TEA gas, so that the Co_3_O_4_-ZnO significantly improves the TEA sensing performance.

## 4. Conclusions

In summary, the Co_3_O_4_-ZnO nanoparticles sensor was successfully fabricated by constructing a heterojunction composite using a simple hydrothermal route and annealing process. The ZnO sensor based on Co_3_O_4_ modification significantly improved the TEA sensing performance of the pure ZnO sensor. Specifically, when the Co content is 1 at%, the Co_3_O_4_-ZnO sensor has higher sensitivity, better selectivity, and faster response recovery speed compared to TEA. The composite sensor also has good repeatability and a low operating temperature, indicating the potential application prospect of the Co_3_O_4_-ZnO composite for TEA detection. In the next study, we will focus on developing other heterojunction composite high performance sensors for gas detection.

## Figures and Tables

**Figure 1 nanomaterials-09-01599-f001:**
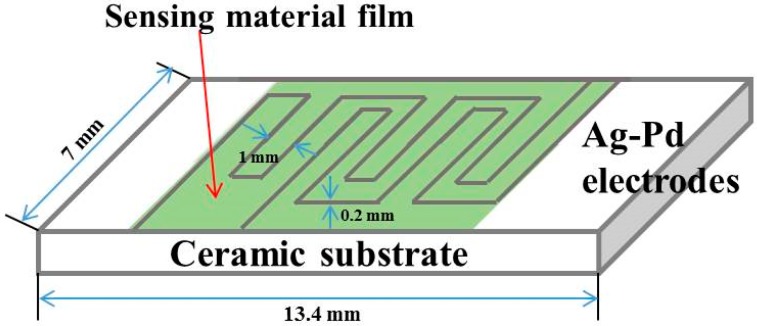
Schematic diagram of a gas sensor manufactured.

**Figure 2 nanomaterials-09-01599-f002:**
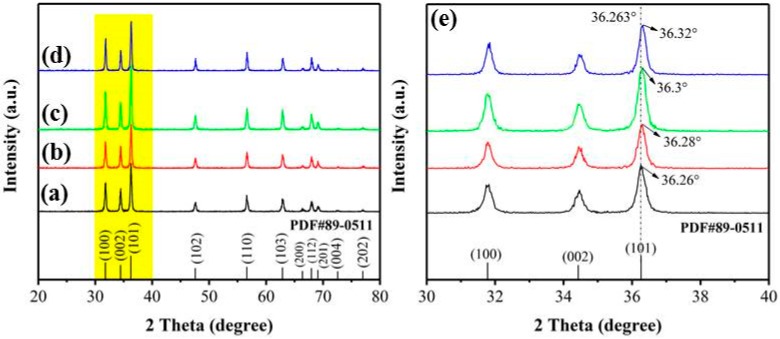
XRD patterns: (**a**) ZnO, (**b**) Co_3_O_4_/ZnO-1, (**c**) Co_3_O_4_/ZnO-2, and (**d**) Co_3_O_4_/ZnO-3; (**e**) the enlarged view of the shaded portion of yellow in (**a**–**d**).

**Figure 3 nanomaterials-09-01599-f003:**
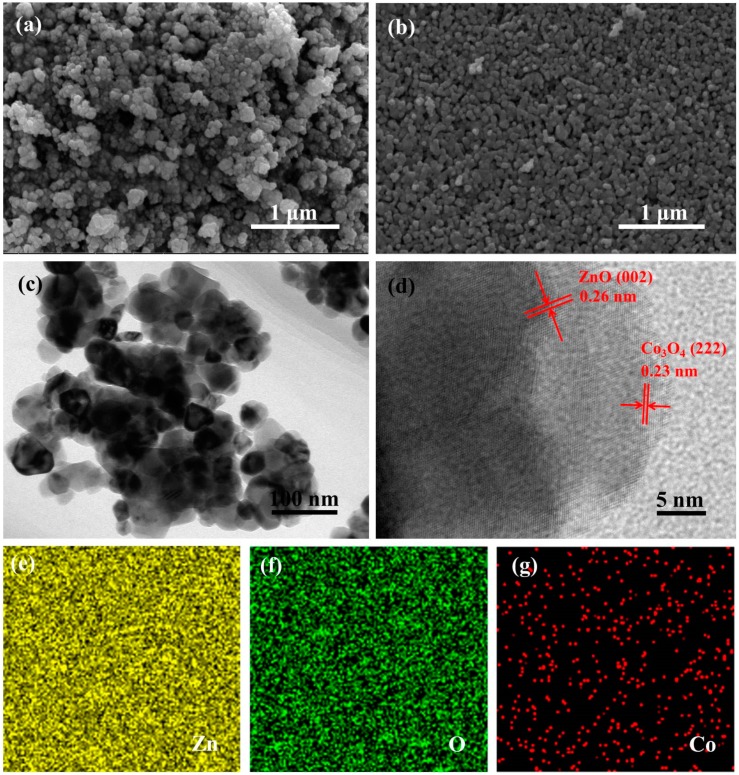
Field emission scanning electron microscopy images of (**a**) ZnO, (**b**) Co_3_O_4_/ZnO-2; (**c**,**d**) transmission electron microscopy and high resolution transmission electron microscope images of Co_3_O_4_/ZnO-2; (**e**–**g**) the energy dispersion spectrum of Co_3_O_4_/ZnO-2 (The two opposite arrows in Figure 3d were used to emphasize the spacing of the two parallel lines as the lattice spacing).

**Figure 4 nanomaterials-09-01599-f004:**
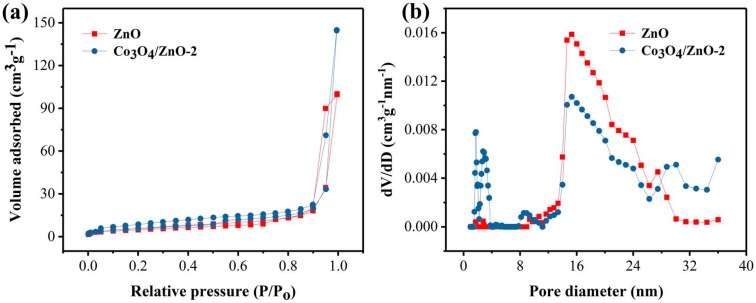
(**a**) N_2_ adsorption–desorption isotherms and (**b**) pore size distribution curves of the prepared ZnO and Co_3_O_4_/ZnO-2 samples.

**Figure 5 nanomaterials-09-01599-f005:**
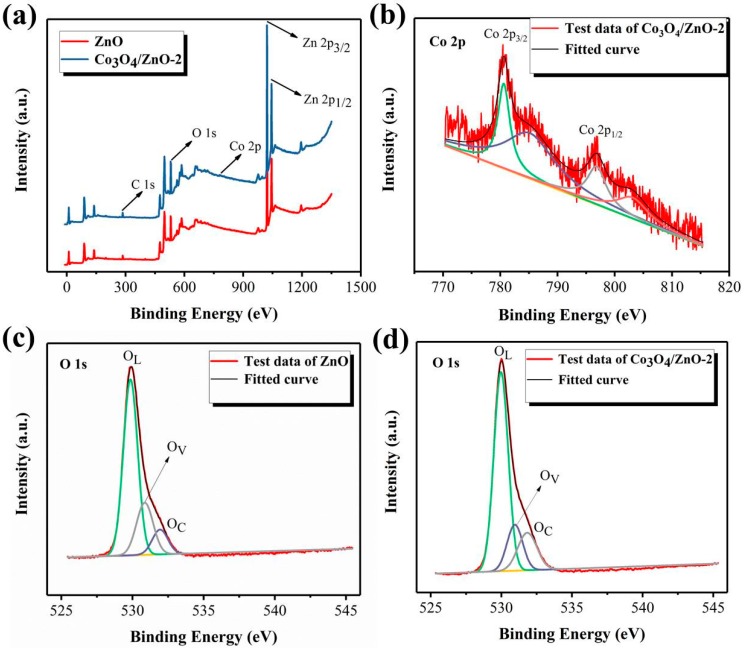
X-ray photoelectron spectroscopy spectra of ZnO and Co_3_O_4_/ZnO-2: (**a**) The full spectra, (**b**) Co 2p spectrum of Co_3_O_4_/ZnO-2, (**c**,**d**) O 1s spectra of the ZnO and Co_3_O_4_/ZnO-2.

**Figure 6 nanomaterials-09-01599-f006:**
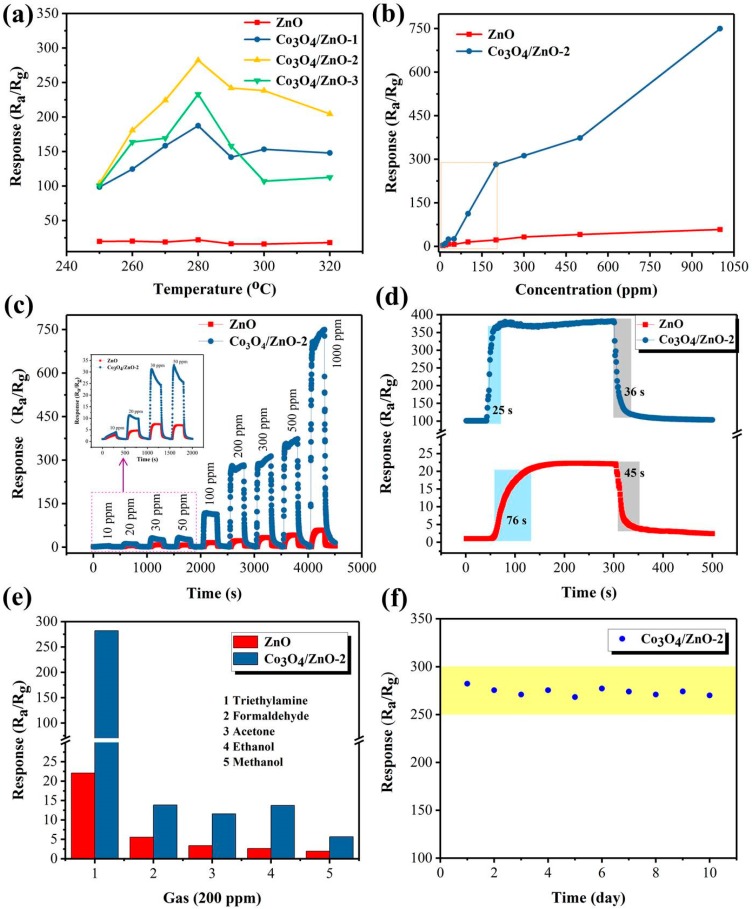
(**a**) Responses of the sensors to 200 ppm triethylamine (TEA) at different temperatures; (**b**) concentration-dependent responses of the sensors toward TEA at 280 °C; (**c**,**d**) dynamic and transient response–recover curves of the sensors toward TEA at 280 °C, where d is 200 ppm TEA; (**e**) the responses of the sensors to 200 ppm various gases at 280 °C; (**f**) stability research of the Co3O4/ZnO-2 sensor to 200 ppm TEA at 280 °C (define R_a_/R_g_ as the response of the sensor).

**Figure 7 nanomaterials-09-01599-f007:**
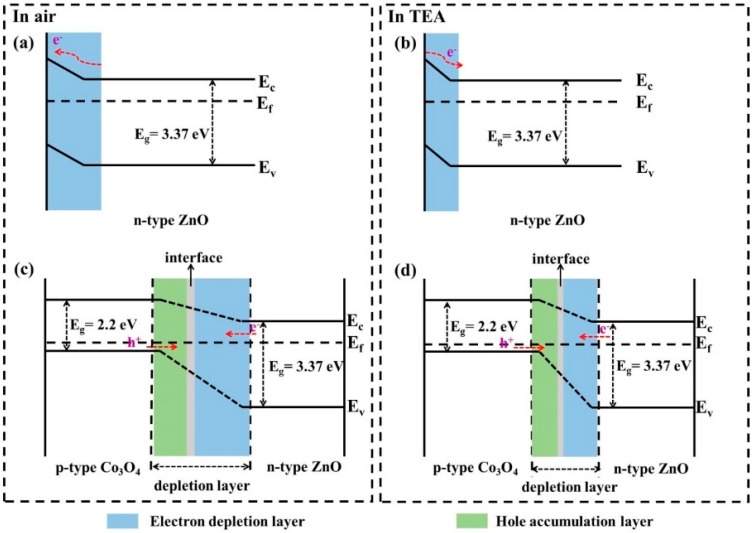
The energy band diagrams of (**a**,**b**) ZnO and (**c**,**d**) Co_3_O_4_-ZnO during exposure to air and TEA, respectively.

**Table 1 nanomaterials-09-01599-t001:** Comparison of the states of O_L_, O_V_, and O_C_ in ZnO and Co_3_O_4_/ZnO-2 (Ene. O_L_/O_V_/O_C_—the binding energy of O_L_/O_V_/O_C_) (O_L_, O_V_, and O_C_ are lattice oxygen, vacancy oxygen and chemisorbed oxygen, respectively).

Materials	O_L_ (%)	O_V_ (%)	O_C_ (%)	Ene. O_L_ (eV)	Ene. O_V_ (eV)	Ene. O_C_ (eV)
ZnO	66.8	22.99	10.21	529.85	530.83	531.93
Co_3_O_4_/ZnO-2	66.6	16.75	16.65	529.97	530.95	531.84

**Table 2 nanomaterials-09-01599-t002:** TEA sensing properties of the sensors based on different materials (temperature, concentration, response, response time/recovery time; references were abbreviated as Tem, Con, Res, T_res_/T_rec_, and Ref, respectively).

Materials	Tem. (°C)	Con. (ppm)	Res.	T_res_/T_rec_. (s)	Ref.
SnO_2_ nanorods	350	50	64.8	-	[29]
NiO-ZnO nanosheets	320	100	300	-	[30]
Nano-CoFe_2_O_4_	190	10	2	100/120	[31]
α-Fe_2_O_3_ microrods	275	100	11.8	-	[32]
TiO_2_-SnO_2_ nanosheets	260	100	52.3	-	[33]
SnO_2_-Zn_2_SnO_4_ spheres	250	100	48	-	[34]
TiO_2_ nanorods	290	100	14.2	-	[35]
V_2_O_5_ hollow spheres	370	100	7.2	20/96	[36]
Au@SnO_2_/MoS_2_	300	100	96	12/66	[37]
Au@ZnO/SnO_2_	300	100	115	7/30	[38]
CeO_2_-SnO_2_ nanoflowers	310	200	252.2	-	[39]
Co_3_O_4_-ZnO	280	100	112.5	22/26	This work
	280	200	282.3	25/36	

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
