# Peer review of "Hydrothermal Synthesis of Co3O4/ZnO Hybrid Nanoparticles for Triethylamine Detection"

_nanomaterials, 2019, doi:10.3390/nano9111599_

Round 1

Reviewer 1 Report

This article details the design, construction, and testing of an interdigitated electrode coated with a layer of Co3O4/ZnO nanoparticles and their ability to measure trimethylamine concentrations in a gaseous flow. Overall, the article is well-written and the methods and results are clearly described. It is recommended that this article be published after some revisions that are detailed below:

Page 2, lines 68-70. Is the purpose of the annealing step to reduce the starting materials to Co3O4 and ZnO? Or is there an additional reason? It is recommended that a short purpose for the annealing step be included. Page 3, Section 2.3: Could more a more detailed description of the interdigitated electrode design be provided? If the electrode wafers are purchased, a manufacturer would be appreciated. If they are constructed in the laboratory, then details such as electrode width, overlap distance, and general preparation procedures would be helpful. Page 3, Figure 1: Is the “Sensing material film” layer misaligned with the substrate? It looks like only a portion of the interdigitated electrode is covered by the film. Page 3, Figure 2: Is this figure truly necessary? In none of the nanoparticle formulations is there any evidence of the Co3O4. Was there a thought that the addition of the Co3O4 would change the structure of the ZnO? Page 5, lines 122-124: The calculated surface area increase by adding the Co3O4 is only about 20%. Is this consistent with the significant change in the overall morphology of the nanoparticles as shown in Figure 3a and 3b? Page 5, lines 138-143. This is not a critique of the results, but it is suggested that the comparison of the OL, OV, and OC states be placed in a table to make it easier to compare. That, however, is a personal opinion and if it is decided not to do this, please disregard. Page 7, Figure 6a: Are there any error bars that could be added to this graph to give some indication of reproducibility? Page 7, Figures 6b and 6 c: Is there some explanation for the general shape of the response curve for the Co3O4/ZnO nanpaticles? It seems that the ZnO nanoparticles show a more systematic trend. Page 7, Figure 6c: Is there any explanation for what seems like desorptive behavior at lower TEA concentrations (less than 100 ppm) where the resistance reaches a peak and then gradually declines? Page 7, Figure 6f: Could you provide an explanation of the shaded region in this figure? Overall conclusions: From the data presented here (most notably in Figure 6), it seems that the minimum detectable amount of TEA is approximately 10 ppm. Can you please comment on the fact that the minimum detectable quantity using the OSH approved method (https://www.osha.gov/dts/sltc/methods/partial/pv2060/2060.html) is 0.04 ppm? It is understood that this is a gas chromatography method that does use some pre-concentrating steps using alumina tubes. It is recognized that the proposed sensor most likely will provide a faster response than the OSHA method, however, its larger minimum detectable quantity can be the main limitation as the current recommended time weighted average (TWA) of TEA is 10 ppm, and a 15 minute TWA is 15 ppm, both approaching the limit of the sensor.

Author Response

Author's Reply to the Review Report (Reviewer 1)

Response: We appreciate your positive comments and constructive suggestions. We carefully checked the manuscript and modified the manuscript based on your comments. All the revisions were highlighted in red in the revised manuscript. The follow is our point‒by‒point reply to your comments:

Page 2, lines 68-70. Is the purpose of the annealing step to reduce the starting materials to Co3O4and ZnO? Or is there an additional reason? It is recommended that a short purpose for the annealing step be included.

Reply: We sincerely thank you for your careful reading and valuable suggestions. The main purpose of the annealing is to obtain the more stable sample to ensure better stability of the gas sensing material during gas sensitivity testing.

Page 3, Section 2.3: Could more a more detailed description of the interdigitated electrode design be provided? If the electrode wafers are purchased, a manufacturer would be appreciated. If they are constructed in the laboratory, then details such as electrode width, overlap distance, and general preparation procedures would be helpful. 

Reply: Thank you for your careful review and valuable comments. Based on your suggestions, we have modified the substrate description in section 2.3 on page 3 of the original manuscript. It should be noted that the substrate with Ag-Pd electrode used in this work was purchased from Beijing Alite Technology Co., Ltd., and we would like to express our sincere gratitude to the manufacturer. Details regarding the substrate have been described in the revised manuscript, as shown in lines 84-87 on page 3 of the revised manuscript and highlighted in red. The specific content is: change ‘’The paste obtained by mixing a small amount of sample and a proper amount of deionized water was brushed on the Ag-Pd electrode region of the ceramic substrate to obtain a gas sensor” to “The paste obtained by mixing a small amount of sample and an appropriate amount of deionized water was brushed on the Ag-Pd electrode region of a ceramic substrate (electrode line width 0.2 mm, pitch 1.0 mm; entire substrate 13.4 mm × 7 mm) purchased from Beijing Elite Technology Co., Ltd. to obtain a gas sensor”.

Page 3, Figure 1: Is the “Sensing material film” layer misaligned with the substrate? It looks like only a portion of the interdigitated electrode is covered by the film

Reply: We apologize for our carelessness. Based on your comment, we modified Figure 1 on page 3 of the original manuscript and displayed it in Figure 1 on page 3 of the revised manuscript. In this work, the paste (sensing material film) of both the sample and the deionized water was uniformly brushed on the Ag-Pd electrode region of the purchased substrate.

Figure 1. Schematic diagram of a gas sensor manufactured.

Page 3, Figure 2: Is this figure truly necessary? In none of the nanoparticle formulations is there any evidence of the Co3O4. Was there a thought that the addition of the Co3O4 would change the structure of the ZnO?

Reply: We sincerely appreciate your valuable comments. Based on your suggestions, we have adjusted Figure 2 on page 3 of the original manuscript. The changed picture is shown in Figure 2 on page 4 of the revised manuscript. Further, as shown in Figure 2, Figure 2f is an enlarged view of a shaded portion of yellow in Figure 2a-d. Figure 2f shows that in the range of 2θ = 30°~40°, 2θ = 36.263° in Figure 2a is very close to the diffraction angle of 2θ = 36.26° corresponding to the (101) crystal plane of ZnO (JCPDS no. 89-0511). Moreover, the peak positions of Figures 2b-d are offset by 0.02°, 0.04°, 0.06°, respectively, compared to Figure 2a. This may be mainly due to the introduction of Co3O4 in the composite. However, the sample with a Co content of 3 at% in Figure 2d still has no obvious characteristic diffraction peak of Co3O4, which may be caused by the low content of Co in the composite sample. The added description of this section is highlighted in red font on lines 112-117 on page 3 of the revised manuscript.

Figure 2. XRD patterns: (a) ZnO, (b) Co3O4/ZnO-1, (c) Co3O4/ZnO-2, and (d) Co3O4/ZnO-3; (f) the enlarged view of the shaded portion of yellow in (a-d).

Page 5, lines 122-124: The calculated surface area increase by adding the Co3O4is only about 20%. Is this consistent with the significant change in the overall morphology of the nanoparticles as shown in Figure 3a and 3b? 

Reply: We are very grateful for your careful review and valuable comments. The reason may be that the Co3O4/ZnO-2 composite sample in Figure 3b shows nanoparticles lumps with smaller average size and denser distribution than the ZnO in Figure 3a, which is still the agglomerate of nanoparticles. Thus, a portion of the surface areas are still embedded in the nanoparticles lumps, resulting in the composite having only a small increase in specific surface area.

Page 5, lines 138-143. This is not a critique of the results, but it is suggested that the comparison of the OL, OV, and OCstates be placed in a table to make it easier to compare. That, however, is a personal opinion and if it is decided not to do this, please disregard.

Reply: Thank you for your positive comment and valuable suggestion of our manuscript. We added Table 1 on page 6 of the revised manuscript to describe the OL, OV and OC states.

Table 1. Comparison of the states of OL, OV and OC in ZnO and Co3O4/ZnO-2 (Abbreviate the binding energy of Ox as ‘Ene. Ox’).

Materials

OL (%)

 OV (%)

OC (%)

Ene. OL (eV)

Ene. OV (eV)

Ene. OC (eV)

ZnO

66.8

22.99

10.21

529.85

530.83

531.93

Co3O4/ZnO-2

66.6

16.75

16.65

529.97

530.95

531.84

Page 7, Figure 6a: Are there any error bars that could be added to this graph to give some indication of reproducibility?

Reply: We are very grateful for your valuable comment. In the temperature dependence response test of this work, we conducted a large number of repetitive tests, and the results showed that the response of the sensor at a certain temperature has only a small degree of fluctuation. Changing the operating temperature, this phenomenon still exists. It was also found that the sensor maintained the rising-maximum-decreasing tendency at various temperatures. Thus, we took the averages of multiple tests as the responses of the sensors in the figure 6a. In addition, we also read the literatures on the temperature dependence response figure of some gas sensors, but did not find content about error bars. We may not be able to add error bars to this image, and we apologize for this. In the next work, we will learn and try this method as much as possible.

Page 7, Figures 6b and 6c: Is there some explanation for the general shape of the response curve for the Co3O4/ZnO nanpaticles? It seems that the ZnO nanoparticles show a more systematic trend.

Reply: We sincerely appreciate the valuable comments. The response of the composite sensor increases rapidly when the concentration of TEA is 10-200 ppm. Subsequently, as the concentration of TEA continued to increase, the rate of increase in the response of the sensor slowed down, possibly due to the saturation of the adsorption of TEA by the gas sensitive material. Thank you for your suggestion. In the following work, we will continue to explore the impact of its factors, and look forward to the development of higher performances TEA gas sensors with high response while still having a good system function relationship to achieve lower concentration quantitative detection capability.

Page 7, Figure 6c: Is there any explanation for what seems like desorptive behavior at lower TEA concentrations (less than 100 ppm) where the resistance reaches a peak and then gradually declines?

Reply: Thanks for your suggestion. When the concentration of TEA is less than 100 ppm, initially, the concentration of TEA may be relatively low for Co3O4/ZnO gas sensing sensor. As the concentration of TEA increases, it can stimulate the chemically adsorbed oxygen ions on the sensing material to react efficiently with the adsorbed active TEA, so that the response of the sensor increases rapidly. Subsequently, there may be a transitional concentration during which the interaction between the active sites on the sensing material and the TEA gas may be comparable. As the concentration of TEA increases, the magnitude of the increase in the response of the sensor becomes smaller. In the future work, it is necessary for us to continue to explore and develop gas sensors that are highly sensitive to lower concentrations of TEA.

Page 7, Figure 6f: Could you provide an explanation of the shaded region in this figure?

Reply: Thanks for your suggestion. The average response of multiple tests in a day is used as the response value of the sensor, and the ratio of the standard deviation of the responses of the sensor to the average value between the 10 days is used as the coefficient of variation. The calculated coefficient of variation is only about 1.42%, indicating that the response of the composite sensor has only a small degree of fluctuation over a long period of time and has good repeatability.

Overall conclusions: From the data presented here (most notably in Figure 6), it seems that the minimum detectable amount of TEA is approximately 10 ppm. Can you please comment on the fact that the minimum detectable quantity using the OSH approved method (https://www.osha.gov/dts/sltc/methods/partial/pv2060/2060.html) is 0.04 ppm? It is understood that this is a gas chromatography method that does use some pre-concentrating steps using alumina tubes. It is recognized that the proposed sensor most likely will provide a faster response than the OSHA method, however, its larger minimum detectable quantity can be the main limitation as the current recommended time weighted average (TWA) of TEA is 10 ppm, and a 15 minute TWA is 15 ppm, both approaching the limit of the sensor. 

Reply: We appreciate your valuable comments. We are very sorry that we did not find the specific content of the URL.

(https://www.osha.gov/dts/sltc/methods/partial/pv2060/2060.html)

According to our understanding, gas chromatography first vaporizes the analytical sample in a vaporization chamber and is then carried into the column (the column contains a liquid or solid stationary phase (alumina, etc)) by an inert gas (carrier gas). Due to the flow of the carrier gas, the sample components are repeatedly dispensed or adsorbed/desorbed during the movement. As a result, a component having a large concentration in the carrier gas flows out of the column first. The sample exiting the column then immediately enters the detector. The detector is capable of converting the presence or absence of a sample component into an electrical signal, which is then amplified by a signal amplifier and finally recorded. In addition, the signal output after the detection will be corrected, which may also provide a certain reference for achieving a lower amount of gas detection.

However, we apologize for the fact that the main limitation of TEA gas sensors based on Co3O4/ZnO materials in this work is that the minimum detectable quantity is larger than the minimum detectable quantity determined by the existing gas chromatography. I hope that in the future work, we will continue to work hard to reduce the minimum gas detection of the sensor.We feel great thanks for your guidance.

Reviewer 2 Report

page 6, line 162 remove "OWT"

page 7, the caption of Figure 6: briefly describe what is "Ra/Rg"

page 7, line 180: describe how "Ra/Rg" has been measured

page 8, line 194, Table 1 caption describe the abbreviations used as names of the columns

Page 9 and page 10 including Conclusions: instead of "Co3O4/ZnO-2" use only "Co3O4/ZnO" - as it is used in Abstract

Author Response

Author's Reply to the Review Report (Reviewer 2)

Response: We appreciate your positive comments and constructive suggestions. We carefully checked the manuscript and modified the manuscript based on your comments. All the revisions were highlighted in red in the revised manuscript. The follow is our point‒by‒point reply to your comments:

page 6, line 162 remove "OWT"

Reply: We sincerely thank you for your careful reading and valuable suggestion. We checked the manuscript and deleted the abbreviation ‘OWT’ for the ‘optimum working temperature’ in the sentence. The abbreviation is not really necessary for the full text. The original sentence "Therefore, 280 °C was determined to be optimum working temperature (OWT) and applied to the following tests" was modified to "Therefore, 280 °C was determined to be optimum working temperature and applied to the following tests". In addition, as shown in lines 184-185 on page 7 of the revised manuscript, the modified sentence is highlighted in red.

page 7, the caption of Figure 6: briefly describe what is "Ra/Rg"

Reply: We sincerely appreciate your valuable comment. As suggested by the reviewer, we have added a brief description of Ra/Rg in the caption of Figure 6. As shown in line 196 of page 8 of the revised manuscript, the added content "Define Ra/Rg as the response of the sensor" has been highlighted in red.

page 7, line 180: describe how "Ra/Rg" has been measured

Reply: Thank you for your valuable and constructive suggestion. Based on your comment, we have added a description of the Ra/Rg measure in the ‘Construction of gas sensors’ section of the original manuscript. The newly added description is shown in lines 92-104 on page 3 of the revised manuscript and highlighted in red. The details are as follows.

In particular, pairs of adjustable probes are in contact with and are attached to both ends of the fabricated sensor. The power supply, water circulation system, temperature control system and data acquisition system are then turned on. Initially in the atmosphere, and when the resistance of the sensor collected on the display is relatively stable, the target gas is introduced into the test chamber after an interval of 20 seconds, after which the test chamber is quickly closed. In the data acquisition system, the relatively stable resistance when the sensor is in the air is set to Ra, Rg is the resistance of the sensor collected by the system in the test gas, and Ra/Rg is selected in the sensor response curve type column. After the target gas stayed in the test chamber for about 240 seconds, the lid on the upper side of the test chamber was opened to release the test gas. Until the sensor's response basically returns to 10% of the maximum response, click to stop collecting data and save the data. In this way, the Ra/Rg value is directly recorded in the data acquisition system on the intelligent sensing analysis system, and finally exported by computer.

page 8, line 194, Table 1 caption describe the abbreviations used as names of the columns

Reply: We are very grateful for your careful review. Based on your instructive comment, we have modified the original manuscript and added the abbreviated names used in the first column of the table to the caption in Table 1. As shown in lines 217-219 on page 9 of the revised manuscript, the font is highlighted in red. The specific content added is: “Temperature, concentration, response, response time/recovery time, references were abbreviated as Tem, Con, Res, Tres/Trec and Ref, respectively”.

Page 9 and page 10 including Conclusions: instead of "Co3O4/ZnO-2" use only "Co3O4/ZnO" - as it is used in Abstract

Reply: Thank you again for your valuable suggestion of our manuscript. We have modified 'Co3O4/ZnO-2' on pages 9-10 in the original manuscript to 'Co3O4/ZnO', as shown by the red font on pages 10-11 of the revised manuscript.

Round 2

Reviewer 1 Report

Thank you for answering my questions and for expanding on the areas that were requested. I recommend this article be published.